# Octanoic Acid-Enrichment Diet Improves Endurance Capacity and Reprograms Mitochondrial Biogenesis in Skeletal Muscle of Mice

**DOI:** 10.3390/nu14132721

**Published:** 2022-06-29

**Authors:** Anouk Charlot, Lucas Morel, Anthony Bringolf, Isabelle Georg, Anne-Laure Charles, Fabienne Goupilleau, Bernard Geny, Joffrey Zoll

**Affiliations:** 1Centre de Recherche de Biomédecine de Strasbourg, UR 3072 Mitochondrie, Stress Oxydant et Protection Musculaire, Université de Strasbourg, 67000 Strasbourg, France; anthony.bringolf@etu.unistra.fr (A.B.); isabelle.georg@unistra.fr (I.G.); anne.laure.charles@unistra.fr (A.-L.C.); goupilleau@unistra.fr (F.G.); bernard.geny@chru-strasbourg.fr (B.G.); 2Department of Infection and Immunity, Luxembourg Institute of Health, 4354 Esch-sur-Alzette, Luxembourg; lucas_hd@hotmail.fr; 3Faculty of Science, Technology and Medicine, University of Luxembourg, 6, Rue-Kalergi, 1359 Luxembourg, Luxembourg; 4Service de Physiologie et d’Explorations Fonctionnelles Respiratoires, Hôpitaux Universitaires de Strasbourg, 67000 Strasbourg, France

**Keywords:** medium chain fatty acid, octanoic acid, endurance, mitochondrial biogenesis, skeletal muscle

## Abstract

Background: Medium Chain Fatty Acids (MCFAs) are a dietary supplement that exhibit interesting properties, due to their smaller molecular size. The acute consumption of MCFAs is expected to enhance exercise performance. However, the short-term effects of MCFAs on endurance performance remains poorly understood. The aim of our study is to evaluate the octanoic acid (C8)-rich diet effect on endurance capacity, and to explore their molecular and cellular effects. Methods: C57BL/6J mice were fed with a chow diet (Control group) or an octanoic acid-rich diet (C8 diet) for 6 weeks. Spontaneous activity, submaximal and maximal exercise tests were carried out to characterize the exercise capacities of the mice. Beta-oxidation and mitochondrial biogenesis pathways were explored in skeletal muscle by RT-qPCR, Western Blot (Quadriceps) and histochemical staining (Gastrocnemius). Results: Mice fed with a C8-rich diet presented a higher spontaneous activity (*p <* 0.05) and endurance capacities (*p <* 0.05) than the control, but no effect on maximal effort was observed. They also presented changes in the skeletal muscle metabolic phenotype, with a higher number of the oxidative fibers, rich in mitochondria. At the molecular level, the C8-diet induced an AMPK activation (*p <* 0.05), associated with a significant increase in PGC1a and CS gene expression and protein levels. Conclusion: Our study provided evidence that C8-enrichment as a food supplementation improves endurance capacities and activates mitochondrial biogenesis pathways leading to higher skeletal muscle oxidative capacities.

## 1. Introduction

Understanding exercise physiology as well as optimal nutritional needs is important both for sports performance as well as the maintenance of health. Indeed, lipid and carbohydrate metabolism have major effects both on the metabolic health and endurance exercise performance [1,2] Skeletal muscles use several substrates (i.e., carbohydrates, lipids, and amino acids) as energy sources and the ratio in which they are used differs with the exercise intensity and the fitness level [3,4]. During endurance exercise, mitochondrial fat oxidation in skeletal muscles generates an important part of the required energy [5,6]. Interestingly, an increase in fat utilization capacity is associated with improved endurance performance [7,8]. Moreover, it is known that well-trained athletes have an increased fat utilization capacity [9,10]. Next to endurance training, it has been shown that diet could also modify the fat utilization capacity [11,12], even if some studies have reported that prolonged consumption of a high-fat diet consisting of long-chain triglycerides, does not improve exercise performance [13,14].

On the other hand, Medium Chain Triglycerides (MCTs) are a food component that is expected to enhance exercise performance [15]. Indeed, it’s a dietary supplement commonly used along with medications for the treatment of metabolic disease. MCTs are triglycerides made of a glycerol backbone and three fatty acids of a 6 to 12 carbon atom aliphatic tail, named medium-chain fatty acids (MCFA). It has been shown that MCFAs play a role in lowering weight, decreasing metabolic syndrome, fat deposition and food intake [16]. Due to their smaller molecular size, MCTs are digested much more rapidly than long chain triglycerides. MCTs are digested into MFCAs, and then, upon absorption, travel directly to the liver via the portal vein, bypassing the lymphatic system. Thus, MCFAs to serve as a ready source of energy and prevents them from accumulating as fat in body tissues. In a first animal study [17], mice feeding with MCT-containing food for a long period demonstrated a significantly longer swimming time than mice that were fed food containing long-chain triglycerides (LCT). In 2018, Wang et al. showed that in condition of high temperature environment, training mice fed with MCTs enhanced running performance compared to mice fed with regular chow diet. Unfortunately, they did not explore the effect of MCT feeding in normal temperature condition [18]. Furthermore, a previous study on humans [13] reported that ingestion of a beverage containing 4.3% MCT, 10% carbohydrate during 2 h of exercise at 60% peak O_2_ uptake (VO_2_) increased the finishing time in a subsequent simulated 40-km cycling time trial, to levels higher than those brought about by the ingestion of a carbohydrate solution alone. 

Thus, some studies seem to demonstrate that MCTs improve endurance capacities, and especially the octanoic acid. On the other hand, the cellular and molecular mechanisms explaining these effects remain poorly understood, especially the effects of C8 on changes in muscle metabolic phenotype, including reprogramming of mitochondrial biogenesis and the fatty acid utilization pathway 

We hypothesized that a C8-enriched diet increases endurance capacity and induced several adaptations at the muscle metabolic phenotype level. Therefore, we investigated the short-term effect of octanoic acid (C8) consumption on maximal and submaximal (endurance) exercise capacities in mice and explored the molecular pathways involved in the effects of C8, particularly the metabolic molecular pathways leading to the activation of mitochondrial biogenesis of skeletal muscle. Here, we provide the evidence that C8 consumption improves endurance capacity in mice and activates mitochondrial biogenesis pathway leading to an increase in skeletal muscle oxidative capacities.

## 2. Materials and Methods

### 2.1. Animals and Diet

16 twelve-week-old male C57BL/6J mice were purchased from ENVIGO (Gannat, France) and housed at ambient temperature (22 ± 2 °C) under a standard 12 h day/night cycle with an *ad libitum* access to tap water and food. As social animals, mice were housed by two in conventional open-top cages with environmental enrichment as cardboard houses, cotton stick, shredded paper, and wooden chew sticks. Animals were divided into 2 groups: (1) Control group fed with a chow diet (consisting of 8.4% fat from soy oil, 19.3% protein and 72.4% carbohydrate, Safe^®^ Diets), and (2) a medium chain Triglyceride (C8) diet (same diet as control group, but 80% of soy oil was replaced by octanoic acid (Neobee 895^®^)) for 6 weeks. The proportion of C8 enrichment in the C8 diet was based on the study of Fushiki et al. (Fushiki et al., 1995). The Neobee 895^®^ comes from the STEPAN company (Northfield, IL, USA). The diet composition is detailed in Table 1. Body weight and food intake were measured once a week throughout the 6 weeks of the experiment. All experiments were performed in accordance with the Guide for the Care and Use of Laboratory Animal experiments were approved by our local ethics committee (CREMEAS, agreement number: 2018042013495170). 

### 2.2. Spontaneous Activity Measurement 

After 4 weeks of diet, mice were placed in a cage equipped with a spontaneous activity wheel (TSE System) for 72 h. Wheels are connected to a computer that records the parameters pertaining to the voluntary exercise of an animal, including total run duration and total distance. Only the data of the last 48 h were analyzed because the first 24 h were considered as habituation period.

### 2.3. Effort and Endurance Exercise Test 

The exercise tests were performed on a treadmill with a slope of +5 °C (Treadmill Control, Letica, Spain) at the 5th week of protocol. Mice were stimulated to run by an electrical grid (0.2 mA, 0.25 Hz frequency). We established exhaustion, and so the end of the test when mice stayed for 5 s on the grid. For the maximal exercise capacity test, the speed was maintained at 40 cm/s for 90 s and then increased by 3 cm/s each 90 s, until exhaustion. For the endurance test, the speed was maintained at 35 cm/s for 30 min, and then increased to 45 cm/s until exhaustion. Blood samples from the tip of the tail were taken immediately at the end of exercise to measure blood lactate using a lactate Scout+ (EKF diagnostics, Cardiff, UK). 

### 2.4. Anatomical Measurements and Histological Stanning 

Mice were anesthetized in a hermetic cage, ventilated with gas mixture of 4% isoflurane (Aerrane, CSP, Cournon, France) and oxygen. Gastrocnemius and quadriceps muscles were snap-frozen for biochemistry or fixed in isopentane for histological analysis. Mice were euthanized by cervical dislocation and exsanguinated. The gastrocnemius tissue was frozen-fixed in OCT and sectioned at −20 °C on a cryostat microtome (10-μm thick, Cryostar NX70, Fisher Scientific, Waltham, MA, USA). For NADH-tetrazolium reductase staining, cryosections were incubated in 0.2 M Tris-HCl pH 7.4, containing 1.5 mM NADH and 1.5 mM nitrobluetetrazolium (NBT) for 15 min at 55 °C. Then, they were washed with three exchanges of deionized H_2_O. Unbound NBT was removed from the sections with three exchanges each of 30, 60 and 90% acetone solutions in increasing and then decreasing concentration. Finally, the sections were washed several times with deionized water and mounted with aqueous mounting medium (Aquatex, SIGMA-ALDRICH, 108635) [19]. 

### 2.5. Study of Mitochondrial Function 

Left gastrocnemius muscle was dissected on ice under a dissecting microscope and permeabilized by incubation under stirring for 30 min at 4 °C in buffer S with saponin. Then fibers were rinsed with agitation for 10 min at 4 °C in the buffer S. Measuring oxygen consumption in cardiac permeabilized fibers was performed using a high-resolution Oxygraphy with a Clark-type electrode (Oxygraph-2k, Oroboros instruments, Innsbruck, Austria). Fibers (3 ± 0.5 mg wet weight) were incubated twice for 5 min with agitation at 4 °C in MirO5 (EGTA 0.5 mM, MgCl_2_ 3 mM, Potassium lactobionate 60 mM, Taurine 20 mM, KH_2_PO_4_ 10 mM, HEPES 20 mM, Sucrose 110 mM, 2 mg/mL BSA). Then fibers were placed in 2.1 mL of MirO5 in the oxygraphic chamber thermostated at 37 °C. Subsequently, Malate (0.1 mM), ADP (5 mM) and Octanoyl-L-carnitine (100 µM) were added. Results were expressed as pmol/sec/mg wet weight.

### 2.6. RNA Isolation, Reverse Transcription, and Real-Time Quantitative PCR 

Gene expression was measured by real-time quantitative PCR. Total quadriceps RNA was isolated using MagMAX mirVana Total RNA Isolation Kit (Applied Biosystems^TM^, Waltham, CA, USA) with the Kingfisher Duo Prime instrument (Fisher Scientific, MA, USA) and was stored at −80 °C. Quantity and purity were measured with Qubit^TM^ RNA Broad Range (BR) and Integrity Quality (IQ) assay kits (Invitrogen^TM^, Carlsbad, CA, USA) using the Invitrogen Qubit 4 Fluorometer (Invitrogen^TM^, CA, USA). cDNA was synthesized from 1 μg of total quadriceps RNA with Maxima H Minus cDNA Synthesis Master Mix (Fisher Scientific, MA, USA). Real-time PCR was performed in triplicate in a total reaction volume of 15 μL using either PowerTrack^TM^ SYBR Green Master Mix (Applied Biosystems, CA, USA) with primers described in Table 2 and measured in automated QuantStudio 3 Real-Time PCR System (Applied Biosystems^TM^, CA, USA). Linear ranges and optimal RNA concentrations for each primer set were previously determined. Each primer sets were designed to span an exon/exon junction to minimize amplification of genomic DNA and obtained from Applied Biosystems. The acidic ribosomal phosphoprotein P0 (36B4) gene was used as housekeeping gene.

### 2.7. Western Blotting and Antibodies

Quadriceps samples were homogenized in 10 volumes of RIPA buffer (50 mM Tris–HCl (pH 7.5), 150 mM NaCl, 1 mM egtazic acid, 1 mM EDTA, 100 mM NaF, 5 mM Na_3_VO_4_, 1% Triton X-100, 1% sodium dodecyl sulfate (SDS), 40 mM β-glycerophosphate, and protease inhibitor mixture (P8340; Sigma–Aldrich)) and then centrifuged at 10,000× *g* for 10 min (4 °C). 60 µg of extracted proteins were loaded into 5–15% SDS-polyacrylamide gels and transferred into nitrocellulose or PVDF membranes (iBlot 2 Dry Blotting System, Invitrogen, Carlsbad, CA, USA). The membranes were blocked for 1 h at room temperature with 50 mM Tris-HCl (pH 7.5), 150 mM NaCl, and 0.1% Tween 20 (TBS-T) containing 5% skimmed milk. The membranes were incubated with the following primary antibodies: anti-citrate synthase (Santa Cruz, TX, USA, Sc-390693, 1:200), anti-PGC-1α1 (Millipore, Massachusetts, USA, AB3242, 1:1000), anti-COX IV (Santa Cruz, TX, USA, Sc-376731, 1:200), anti-total AMPKα (Cell Signaling, #2532, 1:1000) and anti-phospho AMPKα (Thr172, Cell Signaling, #2535, 1:1000). After a night of incubation at 4 °C, the membranes were washed three times with TBS-T and incubated with anti-rabbit (Cell Signaling, MA, USA, 1:4000 #7074S) or anti-mouse (Cell Signaling, Massachusetts, USA, 1:4000 #7076S) secondary antibodies at room temperature for 1 h. The revelation was assessed using Pierce ECL kit (Thermo Fisher Scientific, CA, USA) or SupraSignal Femto kit (Thermo Fisher Scientific, CA, USA), and proteins were visualized by enhanced chemiluminescence (iBright 1500 Imaging System, Invitrogen, CA, USA). ImageJ Software (version 1.8.0) was used for the quantification, and ponceau coloration was used as the loading control [20,21]. 

### 2.8. Statistical Analyses 

Data shown are means ± SEM. Normal distribution of data was verified with a Shapiro-Wilk test and parametric statistical tests were used throughout. Comparisons were made with a Student T-test or Mann-Whitney test, using GraphPad 8^®^ (GraphPad Software, Inc., San Diego, California, USA). Statistical significance is shown as * *p* < 0.05, ** *p* < 0.01 and *** *p* < 0.001. 

## 3. Results

### 3.1. C8 Enrichment Did Not Affect Body Weight

Twelve-week-old C57BL/6J mice were fed with a chow diet with or without the supplementation of C8 for 6 weeks. The data demonstrated that C8 enrichment did not affect body weight (Figure 1A) or food intake (Figure 1B).

### 3.2. C8 Increased Endurance Capacity and Spontaneous Activity

To assess the effect of C8 supplementation on mice exercise performance, we evaluated maximal exercise capacity as well as endurance capacity with a treadmill and spontaneous activity with activity wheels (Figure 2). C8 had no effect on maximal speed, duration, and lactate levels measured during the maximal exercise test. On the contrary, for sub-maximal (i.e., endurance) test, C8 mice presented a higher total distance (+44%, *p <* 0.05), running duration (+40%, *p <* 0.05) and lower lactate blood levels measured after the end of exercise (−44%, *p <* 0.05). Spontaneous wheel activity measurements recorded for 48 h, revealed a higher activity duration (+56%, *p <* 0.05) and total distance (+65%, *p <* 0.05) in C8 group of animals.

### 3.3. C8 Enrichment Increased Mitochondrial Biogenesis 

To understand the molecular pathways involved in the improvement of endurance capacity and spontaneous activity, we explored the metabolic phenotype of skeletal muscle. First, we measured mitochondrial respiration in situ to characterize mitochondrial function in their normal intracellular assembly and position, preserving essential interactions with other organelles. We measured maximal mitochondrial respiration rates in skinned fibers from the superficial part of the gastrocnemius muscle with octanoate substrate (Figure 3). C8 mice presented a tendency to increase the maximal mitochondrial respiration (+45%, *p* = 0.14). 

We performed a histochemical staining of NADH dehydrogenase activity in the gastrocnemius muscle (Figure 4). The coloration revealed a clear increase in the darkly stained fiber in the C8 group in comparison with the control group, suggesting a higher number of oxidative fibers, rich in mitochondria, induced by the C8-enriched diet.

As AMPK is a critical upstream regulator of metabolism, we investigated its role in C8 effects. We showed that the ratio of phosphorylated AMPK/total AMPK, assessed by Western blotting revealed a higher AMPK activation (+166%, *p <* 0.05) (Figure 5) suggesting that C8 consumption activated AMPK by phosphorylation. This activation could lead to a significant increase in oxidative metabolism and mitochondrial biogenesis pathway, so we performed qPCR and Western Blot analyses to confirm it.

The qPCR expression (Figure 6) revealed that C8 enrichment induced no changes in ACADL, ACADM, PFK and PPARα gene expression. On the other hand, C8 mice presented a higher PGC1α (+55%, *p <* 0.05), TFAM (+60%, *p <* 0.05) and CS gene expression (+48%, *p <* 0.05) compared to control group, but no changes in PGC1β and COX4 genes expression. The effect of C8 was confirmed at the protein level by Western Blotting experiments (Figure 7). Indeed, C8 mice presented a higher PCG1α (+35%, *p <* 0.05) and CS (+26.5%, *p <* 0.05) protein levels. These results suggest that C8 enrichment induces skeletal muscle mitochondrial biogenesis.

## 4. Discussion

We demonstrated that 6 weeks of C8-enrichment diet improves the endurance capacity and spontaneous activity in mice. These ameliorations seem to be the consequence of substantial adaptations at the level of skeletal muscle metabolic phenotype. Indeed, we observed a significant activation of AMPK signaling pathways that should trigger the mitochondrial biogenesis mechanism, as seen by higher PGC1α, TFAM and CS expressions in skeletal muscle. 

In exercise physiology, activities can be separated into two categories: low load- high repetition exercises (as submaximal, endurance efforts), or high load-low repetition exercises (as resistance training or maximal effort) [22]. As ATP is critical for skeletal muscle contractile activity, energetic substrate availability plays a key role in effort performance. 

During maximal incremental test to exhaustion, oxidation of carbohydrates, particularly from glycogen muscle storage, is privileged and above all, is not considered as the major limiting factor during maximal effort, and so could explain the absence of significant effect of MCFA enrichment that we observed for the maximal test [23]. This is in accordance with the similar high lactate concentration found in the two groups showing the preponderant use of glycolysis pathway during this test [24]. On the other hand, oxidative metabolism dominates during the submaximal endurance tests and the substrates availability is a critical factor to maintain the same velocity for the longest time possible.

That’s why more and more evidence support the role of dietary strategies in enhancing endurance performance. As carbohydrate metabolism is the first metabolic pathway of ATP production, carbohydrate consumption before and during exercise is often recommended to improve performance. Unfortunately, this strategy is efficient for events lasting approximately 1 h only, because in longer efforts, the carbohydrate use is limited by both liver and muscle glycogen reserves as well as by the small intestine’s capacity to absorb carbohydrate [25]. Moreover, the high carbohydrate strategy inhibits fatty acid oxidation during exercise, which is not optimal for endurance [11]. The consumption of lipids as an ergogenic aid during endurance exercise seems to be efficient in endurance and ultra-endurance exercise. The ketogenic diet (KD) is a high fat and low carbohydrate diet which is initially used for epilepsy treatment and for metabolic disease management [26]. KD has now become popular among endurance athletes, because of the metabolic adaptations due to the KD intake. KD increases fatty acid oxidation in muscle and ketogenesis [27]. 

However, KD metabolic effects are still controversial. This sort of diet is very restrictive and hard to maintain. Another alternative to KD could be a fat supplementation, without carbohydrate restriction [28]. 

Several studies showed that a supplementation of Omega-3 fatty acids could help athletes, by improving performance and enhancing recovery [29]. 

However, the qualitative aspect of the fatty acids appears fundamental to provide beneficial effects for endurance. During digestion, fatty acids are absorbed by the small intestine. Long chain fatty acids are esterified into triglycerides and incorporated into lipoprotein structures called chylomicrons. Then, chylomicrons enter the lymphatic system before being released into the bloodstream via the jugular vein in the neck, where they will be depleted from most of their triglycerides [30,31]. Because ingested long chain fatty acids reach the circulation mostly in chylomicrons, they are not a very important energy source during exercise, and so a supplementation in long chain fatty acid have little or no effect on exercise metabolism or performance [28]. In contrast, medium chain fatty acids (MCFA), thanks to their short carbon chain, are quickly absorbed into the small intestinal cells and transferred directly into the portal vein to be deliver to the liver for hepatic metabolism [31]. Thanks to this property, MCT has been reported to play a role in reducing body [32]. In our study, we did not find a significant difference between groups in body weight, and our results are consistent with the results of Wang et al. and so, where the mice fed with MCTs presented no significant weight loss compared to control. Our hypothesis is that the enrichment of C8 is not enough to induce a significant impact on weight. Indeed, in the study of Montgomery et al., the mice were fed with a high-fat diet with 45% of the calories from MCFAs, while in our case, the proportion of MFCA was almost six times lower [32]. Moreover, it has been demonstrated that high energy levels of MCFAs is required to achieve weight loss [33]. 

Moreover, MCFAs enter directly into the cells as well as mitochondria, without the need of transporters. Therefore, MCFAs are easily stored in skeletal muscles and could be rapidly oxidized compared to long chain fatty acids and so act as a quick energy supply [34]. This is in line with the suggestion that the consumption of MCFAs increases the energy supply from fat oxidation in addition to the glycogen, and thus improve endurance capacity [35]. The decrease in lactate concentration at the end to the submaximal exercise in the C8 group is also an indicative of the preferential oxidation of fatty acids as energy source, rather than anaerobic glycolysis [36]. A better use of fatty acid is beneficial for endurance because beta-oxidation provides more ATP than glycolysis. It delays the glycogen stock depletion, hypoglycemia, lactate production and so permits sustained endurance efforts by decreasing muscular fatigue [3,37].

This decrease in muscular fatigue could explain the increase in spontaneous activity by reducing their recovery time and then increasing the number of exercise sessions in the wheel. To elucidate if the increase in endurance capacity by C8 could be a result of increased mitochondrial oxidative capacities, we evaluated the levels of mRNA and proteins involved in mitochondrial biogenesis and function. Thus, we characterized metabolic muscular phenotype in these mice by exploring mitochondrial respiration, metabolic fiber types as well as CS protein. C8 mice presented a skeletal muscle mitochondrial respiration increase by 45% (non-significant) with C8 substrates. The histochemical staining of NADH dehydrogenase activity revealed a change in fiber type in C8 mice, where an increase in the number of fibers with high NADH activity suggests that C8-enriched diet induced a shift from glycolytic to oxidative fibers. CS is a classical marker of mitochondria [20,38,39] and then, its increase at the mRNA and protein levels strengthen the augmentation of skeletal muscle mitochondria content in the C8 group [40]. This higher number of mitochondria in skeletal muscle should participate to the amelioration of the endurance capacity of C8 mice, because a higher number of mitochondria means a greater oxidative capacity to produce more ATP and so energy. In the skeletal muscle, the increase in mitochondria induces a change in fiber type. Low mitochondrial content characterizes glycolytic fibers, which have a low resistance to fatigue, whereas oxidative fibers are rich in mitochondrial content and have a high resistance to fatigue. These properties differentiate the respective utility of each fiber type during exercise. Indeed, a greater proportion of oxidative fibers predicts success in slower, longer distance effort (as endurance) and a greater number of glycolytic fibers predict success in higher velocity, shorter duration events (as maximal and supramaximal efforts) [41]. The question now is to better understand how, at the molecular level, a C8-enriched diet could improve muscle oxidative capacity. For this, we studied the level of expression of the main actors that control mitochondrial biogenesis.

AMPK is known as a powerful kinase which acts as a central regulator of energy homeostasis. This protein is involved in the phosphorylation of over 100 distinct proteins spread in diverse array of metabolic pathways, as lipid, carbohydrate, amino acid metabolism and also mitochondrial function, autophagy and cell growth [42]. 

In skeletal muscle, AMPK activation could be link to the increase in free fatty acid availability mediated by the C8 supplementation. Watt et al. demonstrated in L6 myotubes that increasing fatty acid availability increased AMPK activity [43]. Takikawa et al. also confirmed that MCFAs increased AMPK activity in L6 myotubes [44]. In our study, we showed an increase in AMPK activation as seen in the cellular model. This is an important result because when AMPK is activated, it phosphorylates Acetyl-CoA carboxylase 1 (ACC1) in the cytoplasm and Acetyl-CoA carboxylase 2 (ACC2) in the mitochondria, leading to the suppression of fatty acid synthesis and promoting fatty acid oxidation [45]. 

The role of AMPK in endurance was also demonstrated by Narkar et al. with a pharmacological activation of AMPK with agonists, such as 5-amino-4-imidazolecarboxamide ribotide (AICAR), that increased running endurance in untrained mice [46]. In humans, the use of AICAR to improve endurance is prohibited by the World Anti-doping Agency. 

Importantly, AMPK is known to increase PGC1α, the major molecular actor of mitochondrial biogenesis activation [47,48]. It is known that PGC1α activates nuclear respiratory factors 1 and 2 (NRFs) transcription, leading to the production of NRFs proteins. These proteins are recognized by specific sites on the promoter of the mitochondrial transcription factor A (TFAM) and induced its transcription. Finally, TFAM joins the mitochondria and induces mitochondrial DNA transcription as well as its replication [49]. 

Our experiments showed that PGC1α was increased at the mRNA and protein level as well as TFAM at the mRNA level, suggesting that the mitochondrial biogenesis pathway was triggered following the AMPK activation. A significant increase in PGC1α and TFAM were also found in mice supplemented with MCT mix under high temperatures [50]. In the study of Narkar et al., the activation of AMPK with AICAR induced an augmentation of PGC1α gene expression [46]. Altogether, our results suggest that AMPK activation leads to an improvement of endurance capacities in part due to an augmentation of the skeletal muscle oxidative capacities following the triggering of mitochondrial biogenesis pathway. 

The activation of AMPK by the C8 consumption could be explained by the intervention of the Fibroblast Growth Factor 21 (FGF21). This endocrine hormone, primarily expressed in the liver, is involved in cell metabolism by stimulating glucose uptake, fatty acid metabolism and mitochondrial biogenesis, following activation of AMPK by the way of a kinase cascade activation involving the liver kinase B1 [51,52,53]. Loyd et al. demonstrated that FGF21 knockout mice exhibited an impaired adaptation to exercise training mediated by a reduced AMP-activated protein kinase activity in skeletal muscle, suggesting the importance of FGF21 to phosphorylate AMPK [54]. In myotubes, FGF21 deficiency significantly decreased AMPK phosphorylation while the myotubes incubation with FGF21 resulted in a dose-dependent increase in AMPK phosphorylation [55]. FGF21 gene expression is regulated by PPARα fixation in the peroxisome proliferator response elements located in the FGF21 promoter [56]. C8 could increase FGF21 production by PPARs activation or by the production of α-Lipoic acid from octanoic acid, which induces hepatic FGF21 expression [57,58]. However, this hypothesis is based on indirect evidence, and more investigation should be performed to understand the role of FGF21 in AMPK muscle activation. 

In conclusion, our study provided evidence that C8-enrichment, as a food supplementation, improves endurance capacity in mice. This amelioration could be linked to an optimization of the use of energy substrates during exercise and in particular from medium-chain fatty acids. Indeed, we described several muscular adaptations, which should be mediated through the activation of the AMPK-PGC1α-TFAM pathway (Figure 8). These changes lead to an increase in the mitochondria amount associated with a greater oxidative capacity to produce ATP, and results to a switch from glycolytic to oxidative fiber types. These results could give new perspectives in other metabolic pathologies, as obesity or type 2 diabetes, where a decrease in mitochondrial dysfunction, as a decrease in oxidative capacities and mitochondrial biogenesis occurs [59]. 

## Figures and Tables

**Figure 1 nutrients-14-02721-f001:**
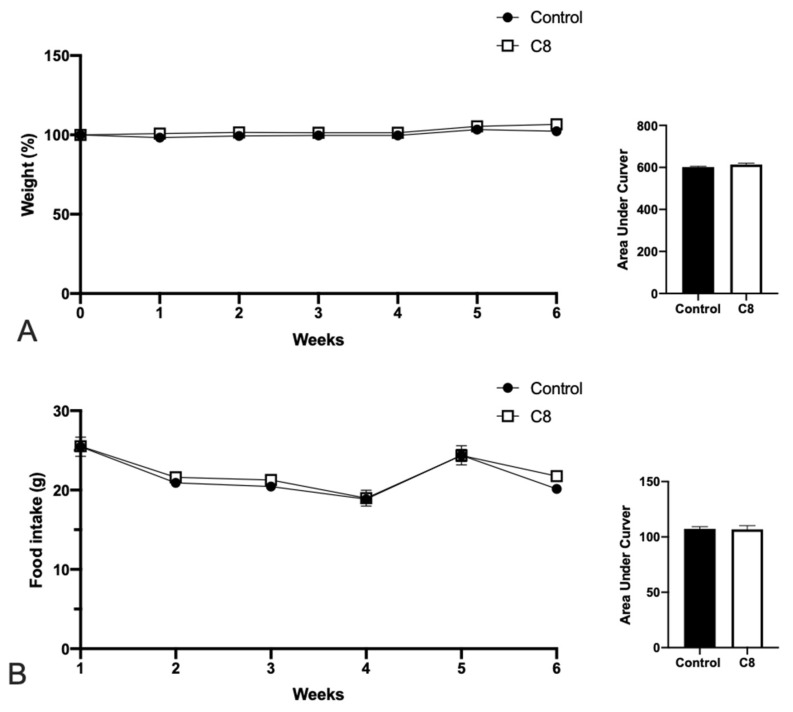
Effect of C8 enrichment on weight gain (**A**) and food intake (**B**). Body weight is expressed in %, where the initial weight of mice was used as 100%, and the weight gain was related to this value. Food intake was expressed in g/week.

**Figure 2 nutrients-14-02721-f002:**
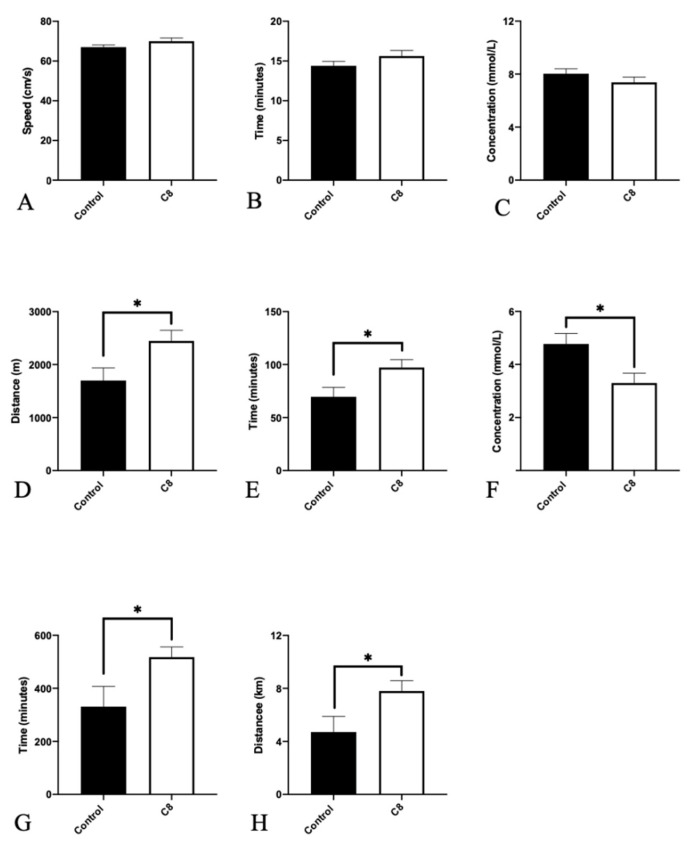
Effect of C8 enrichment on maximal capacity, endurance capacity and spontaneous activity. For maximal exercise capacity evaluation, the maximal speed (**A**), the effort duration (**B**) and the blood lactate levels (**C**) were measured. For Endurance capacity evaluation, the distance (**D**), the duration (**E**) and the blood lactate levels (**F**) were measured. Spontaneous wheel activity was assessed during 48 h by the recording of total duration (**G**) and total distance (**H**) run in the wheel. * *p <* 0.05; n = 6–8.

**Figure 3 nutrients-14-02721-f003:**
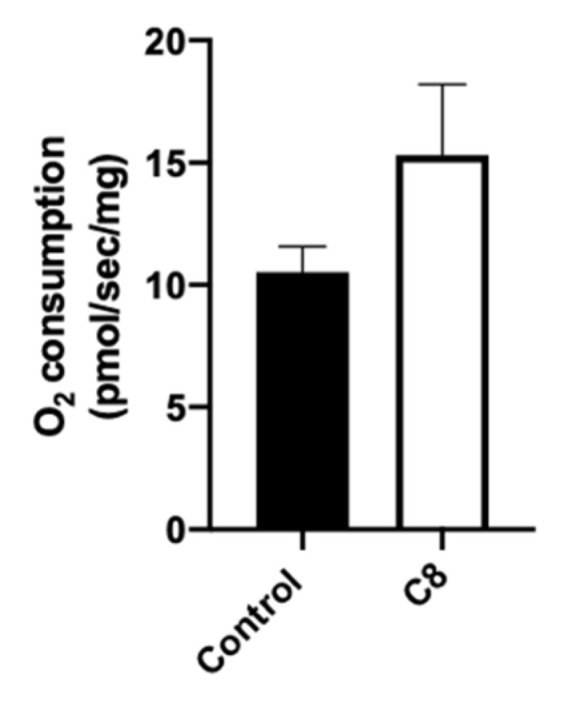
Effect of C8 enrichment on mitochondrial respiration. Maximal mitochondrial respiration was measured after the addition of ADP and 100 µM C8 in gastrocnemius. n = 6–8.

**Figure 4 nutrients-14-02721-f004:**
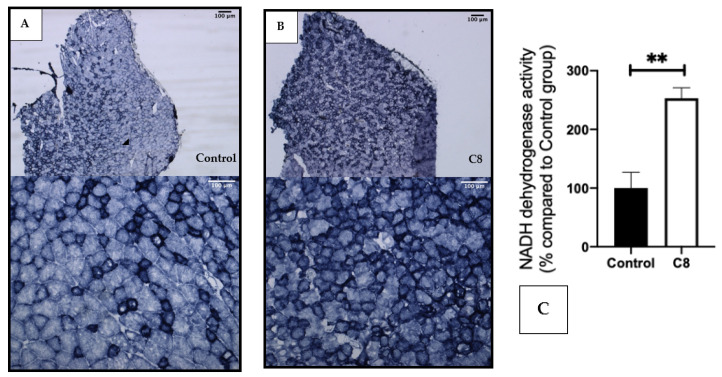
Effect of C8 enrichment on NADH dehydrogenase activity. Histochemical staining of NADH dehydrogenase activity in control gastrocnemius (**A**) and C8 mice (**B**), and its quantification (**C**). Two different fiber-types are distinguished: oxidative fibers are darkly stained; glycolytic fibers are moderately unstained. ** *p <* 0.01; n = 3.

**Figure 5 nutrients-14-02721-f005:**
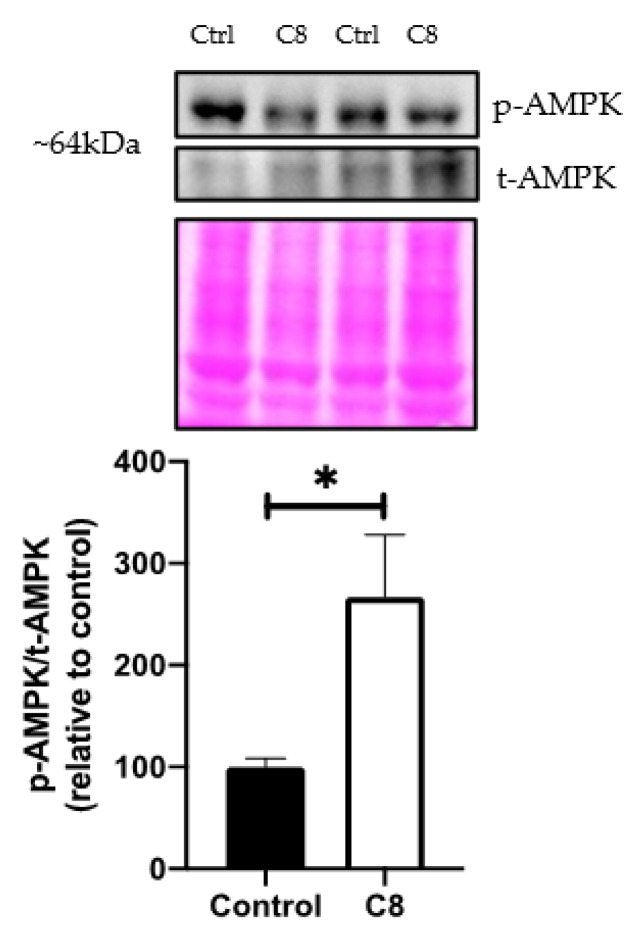
Effect of C8 enrichment on quadriceps AMPK activation. AMPK, AMP-activated protein kinase. Above each panel, western blots from two representative samples are displayed. * *p <* 0.05; n = 7.

**Figure 6 nutrients-14-02721-f006:**
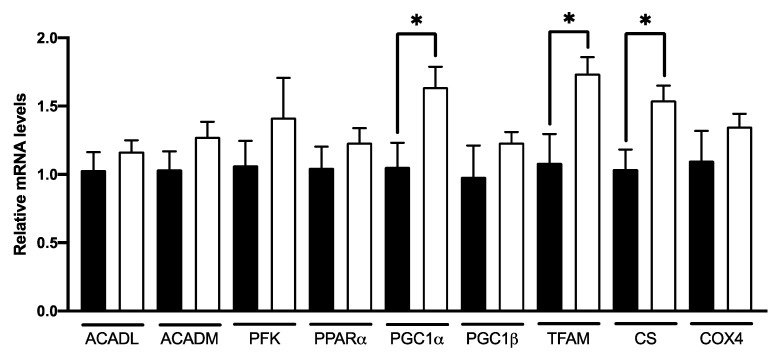
Effect of C8 enrichment on qPCR expression in mice quadriceps. ACADL, acyl-CoA dehydrogenase long chain; ACADM, acyl-CoA dehydrogenase medium chain; PFK, phosphofructokinase; PPARα, peroxisome proliferator activated receptor alpha; PGC1α, peroxisome proliferator-activated receptor gamma coactivator 1-alpha; PGC1β, peroxisome proliferator-activated receptor gamma coactivator 1-beta; TFAM, transcription factor A mitochondrial; CS, citrate synthase; COX4, cytochrome c oxidase subunit 4 mitochondrial. * *p <* 0.05; n = 6–8.

**Figure 7 nutrients-14-02721-f007:**
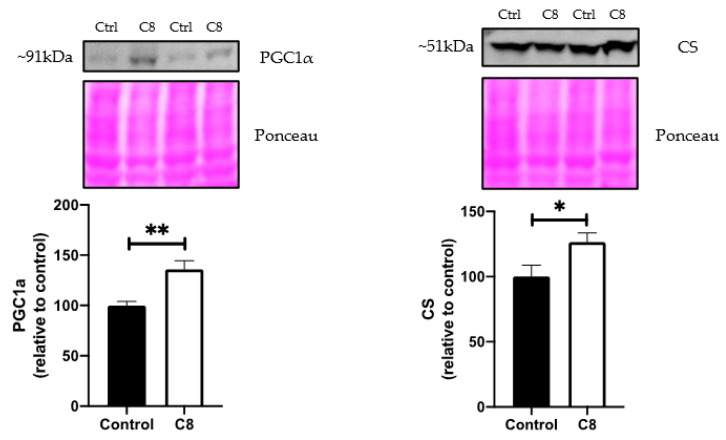
Effect of C8 enrichment on quadriceps protein levels. CS, citrate synthase; PGC1α, peroxisome proliferator-activated receptor gamma coactivator 1-alpha. Above each panel, western blots from two representative subjects are displayed. * *p <* 0.05; ** *p <* 0.01; n = 7.

**Figure 8 nutrients-14-02721-f008:**
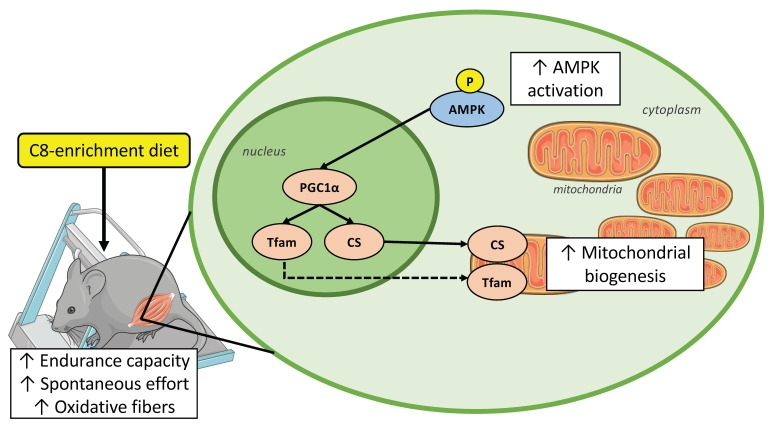
C8-enrichment diet improves endurance capacity by engaging the AMPK-PGC1α-TFAM pathway which lead to improved fatty acid oxidation capacities in skeletal muscle of mice. Full-arrow means direct action on protein, dotted line arrow means direct action on gene expression. ↑ increase.

**Table 1 nutrients-14-02721-t001:** Diet composition.

Diet Composition (g/kg)	Control Diet	C8 Diet
Acid Casein	200.0	200.0
Corn starch	367.5	367.5
Maltodextrin	132.0	132.0
Sugar	100.0	100.0
Cellulose (Arbocel B600)	50.0	50.0
Soya oil	100.0	20.0
Caprylic acid (Neobee 895^®^)	-	80.0
Vitamin mix AIN-93	10.0	10.0
Mineral mix AIN-93G	35.0	35.0
L-Cystine	3.0	3.0
Choline bitartrate	2.5	2.5
Tert.butyl hydroquinone	0.014	0.014
Total	1000.0	1000.0

**Table 2 nutrients-14-02721-t002:** Primers for quadriceps qPCR.

Gene	Forward Primer	Reverse Primer
36B4	GAGGAATCAGATGAGGATATGGGA	AAGCAGGCTGACTTGGTTGC
ACADL	GAAGATGTCCGATTGCCAGC	AGTTTATGCTGCACCGTCTGT
ACADM	ATGACAAAAGCGGGGAGTACC	CCATACGCCAACTCTTCGGT
PFK	GGTTTGGAAGCCTCTCCTCC	GCAGCATTCATACCTTGGGC
PGC1a	GACCGCTTTGAAGTTTTTGG	AGCAGGGTCAAAATCGTCTG
PGC1b	AGATTGTAGAGTGCCAGGTGCTGA	TGCTCTGAACACCGGAAGGTGATA
PPARa	ACTACGGAGTTCACGCATGTG	TTGTCGTACACCAGCTTCAGC
TFAM	CCGTATTGCGTGAGACGAAC	TGAAAGTTTTGCATCTGGGTGT
COX4	GTCTTGGTCTTCCGGTTGCG	TTCACAACACTCCCATGTGCT
CS	CGGTTTGTCTACCCTTCCCC	GGCAGGATGAGTTCTTGGCT

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
