# Peer review of "Octanoic Acid-Enrichment Diet Improves Endurance Capacity and Reprograms Mitochondrial Biogenesis in Skeletal Muscle of Mice"

_nutrients, 2022, doi:10.3390/nu14132721_

Round 1

Reviewer 1 Report

The rationale for the study is potentially relevant based on the potential glycogen sparing effects of MCFAs supplementation. 

Abstract

The authors state that the acute effects of MCFAs are expected to enhance exercise performance, but later in the Introduction present no prior studies (and there are none to my knowledge) that show that acute exercise performance is enhanced by MCFAs.

In the abstract and throughout presentation of the Results, the authors use the term increased when the statistical comparisons are being made at endpoint in a comparison of CON vs. C8. In this case, the terms "higher" or "lower" should be used.

Introduction

It is unclear why the authors include anything about health-related issues in the introduction along with exercise performance. It does not fit and should not be included. 

Be consistent with MCTs vs. MCFAs

In lines, 60-69, how did you draw conclusions from the work cited. Your work really focused on characterizing potential molecular mechanisms.

You describe your study as chronic, but this would be considered short-term

Methods: 

Much more information is needed regarding housing conditions of the mice (e.g. cage type, bedding material, etc).

How did you come to use this concentration of C8 in the diet? A rationale and any supporting details of preliminary studies or previous publications are needed.

The imaging instrumentation is not needed nor the imaging magnification, etc.

Why did you choose did levels of statistical acceptance? Was this done a priori? You do not describe this as an exploratory analysis.

Results:

You need to state in Fig 4 that these are representative. Another major flaw is the lack of quantification which could be conducted using Image J. 

Western blot data is not rigorously presented with only one sample shown per group and in some blots without loading controls

Author Response

The rationale for the study is potentially relevant based on the potential glycogen sparing effects of MCFAs supplementation. 

Abstract

The authors state that the acute effects of MCFAs are expected to enhance exercise performance, but later in the Introduction present no prior studies (and there are none to my knowledge) that show that acute exercise performance is enhanced by MCFAs.

In the abstract and throughout presentation of the Results, the authors use the term increased when the statistical comparisons are being made at endpoint in a comparison of CON vs. C8. In this case, the terms "higher" or "lower" should be used.

à Thank you for the comment, the modification of the terms has been changed in the abstract and results.

Introduction

It is unclear why the authors include anything about health-related issues in the introduction along with exercise performance. It does not fit and should not be included. 

à This part was removed from the introduction.

Be consistent with MCTs vs. MCFAs

à Line 64-67: We added some precisions to make the proper difference between MCT and MCFA.

In lines, 60-69, how did you draw conclusions from the work cited. Your work really focused on characterizing potential molecular mechanisms.

à Thank you for the comment, we modified this section in order to be clearer.

You describe your study as chronic, but this would be considered short-term.

à Thank you for your comment, the term “Chronic” was modified by “Short-term”

Methods: 

Much more information is needed regarding housing conditions of the mice (e.g. cage type, bedding material, etc).

à More information about the housing conditions have been added line 91-95

How did you come to use this concentration of C8 in the diet? A rationale and any supporting details of preliminary studies or previous publications are needed.

à Thank you for this comment. Octanoic acid was chosen after preliminary experiments where we tested the effect of C8, C10 and a mix C8/C10 on endurance. C10 and the mix C8/C10 did not induced a significant effect on endurance, instead of the C8.

Moreover, the proportion of C8 in the food was based on the article of Fushiki et al. The enrichment of C8 can’t be 100%, because mice need to have an apport in LCFA around 20%. (https://pubmed.ncbi.nlm.nih.gov/7876928/). It has been added in material and methods.

The imaging instrumentation is not needed nor the imaging magnification, etc

à We did not quite understand this remark. We did not put the imaging instrumentation, but we prefer to leave the magnitude on the figure which should be important for readers.

Why did you choose did levels of statistical acceptance? Was this done a priori? You do not describe this as an exploratory analysis.

à We used the level of acceptance and the statistical tests conventionally used in all the studies on this research field (https://pubmed.ncbi.nlm.nih.gov/7876928/; https://pubmed.ncbi.nlm.nih.gov/29420554/)

Results:

You need to state in Fig 4 that these are representative. Another major flaw is the lack of quantification which could be conducted using Image J. 

à The quantification has been conducted and added at the Figure 4.

Western blot data is not rigorously presented with only one sample shown per group and in some blots without loading controls

à The WB images have been improved accordingly. 

Reviewer 2 Report

Medium-chain triglycerides/fatty acids are generally considered a good biologically inert source of energy and have potentially beneficial attributes in protein metabolism. MCTs easily diffuse from the GI tract to the blood without requirement of modification like long-chain or very-long-chain fatty acids. In this presentation, Charlot and colleagues investigated endurance capacity and mitochondrial biogenesis in skeletal muscle of mice by providing octanoic acid rich diet, which is valuable for sports performance and maintenance of health. I have few minor comments that must be addressed.

Authors mentioned they have hypothesized that C8-rich diet increases endurance capacity and induced several adaptations at the muscle metabolic phenotype level. Did authors come to this hypothesize based on any preliminary data testing C8 along with C6 and C10? Also, please include a statement why do you choose C8 over other octanoic acid.

There are several literature that shows C8 rich diet reduce body weight. No such observation in this manuscript. Do you have any comment to add on the manuscript?

Authors have quantified the expression of CS that represents mitochondrial oxidative capacity. Higher expression of CS is associated with higher mitochondrial function and doesn’t mean higher number of mitochondria or mitochondrial amount in the skeletal muscle.

Authors showed increased activity of AMPK in C8 enriched model. It could be more supportive for the oxidative capacity of the mitochondria if they have included an experiment to measure ACC phosphorylation in C8 positive and negative model.

Keep using scientific format of et al throughout the manuscript. Correct Watt and al, Takikawa and al, and Loyd and al as watt et al and so.

Sentences, “To understand the way involved in the improvement of endurance capacity and spontaneous activity, we explored the metabolic phenotype of skeletal muscle” and “As carbohydrate metabolism is the first way of ATP production, carbohydrate consumption before and during exercise is often recommended to improve performance” are not clear. What is meant by way? Could you please use appropriate word?

Provide proper label for WB images in figure 7.

There is no housekeeping gene as a control for WB experiments. What is relative to control and how do you calculate it? Since t-AMPK expression is higher in C8- rich model, we can’t use t-AMPK as a control.

Author Response

Medium-chain triglycerides/fatty acids are generally considered a good biologically inert source of energy and have potentially beneficial attributes in protein metabolism. MCTs easily diffuse from the GI tract to the blood without requirement of modification like long-chain or very-long-chain fatty acids. In this presentation, Charlot and colleagues investigated endurance capacity and mitochondrial biogenesis in skeletal muscle of mice by providing octanoic acid rich diet, which is valuable for sports performance and maintenance of health. I have few minor comments that must be addressed.

Authors mentioned they have hypothesized that C8-rich diet increases endurance capacity and induced several adaptations at the muscle metabolic phenotype level. Did authors come to this hypothesize based on any preliminary data testing C8 along with C6 and C10? Also, please include a statement why do you choose C8 over other octanoic acid.

à Thank you for this comment. Octanoic acid was chosen after preliminary experiments where we tested the effect of C8, C10 and a mix C8/C10 on endurance. C10 and the mix C8/C10 did not induced a significant effect on endurance, instead of the C8.

Moreover, the proportion of C8 in the food was based on the article of Fushiki et al and so. The enrichment of C8 can’t be 100%, because mice need a supply of various fatty acids that are present in soya oil  at a minimum of 20%. (https://pubmed.ncbi.nlm.nih.gov/7876928/). It has been added in material and methods.

There are several literature that shows C8 rich diet reduce body weight. No such observation in this manuscript. Do you have any comment to add on the manuscript?

à Thank you for this comment. We added a paragraph line 418-427 with a hypothesis to explain the absence of C8 effect on weight loss in our study, based on literature.

Authors have quantified the expression of CS that represents mitochondrial oxidative capacity. Higher expression of CS is associated with higher mitochondrial function and doesn’t mean higher number of mitochondria or mitochondrial amount in the skeletal muscle.

à Thank you for your comment. Several studies suggest that citrate synthase protein content, and not only its activity, also appears to be related to this parameter (Shang et al, 2019; Lepsen et al, 2016; Galpin et al, 2012; Arc-Chagnaud et al, 2020; Mallard et al, 2022). We added some of these references to support the use of CS protein levels to reflects mitochondrial content.

Shang H, Xia Z, Bai S, Zhang HE, Gu B, Wang R. Downhill Running Acutely Elicits Mitophagy in Rat Soleus Muscle. Med Sci Sports Exerc. 2019 Jul;51(7):1396-1403. doi: 10.1249/MSS.0000000000001906. PMID: 30649103.

Iepsen UW, Munch GD, Rugbjerg M, Rinnov AR, Zacho M, Mortensen SP, Secher NH, Ringbaek T, Pedersen BK, Hellsten Y, Lange P, Thaning P. Effect of endurance versus resistance training on quadriceps muscle dysfunction in COPD: a pilot study. Int J Chron Obstruct Pulmon Dis. 2016 Oct 27;11:2659-2669. doi: 10.2147/COPD.S114351. PMID: 27822028; PMCID: PMC5087783.

Galpin AJ, Raue U, Jemiolo B, Trappe TA, Harber MP, Minchev K, Trappe S. Human skeletal muscle fiber type specific protein content. Anal Biochem. 2012 Jun 15;425(2):175-82. doi: 
10.1016/j.ab.2012.03.018. Epub 2012 Mar 30. PMID: 22469996; PMCID: PMC3358799.

Arc-Chagnaud C, Py G, Fovet T, et al. Evaluation of an Antioxidant and Anti-inflammatory Cocktail Against Human Hypoactivity-Induced Skeletal Muscle Deconditioning. Front Physiol. 2020;11:71. Published 2020 Feb 12. doi:10.3389/fphys.2020.00071

Mallard J, Hucteau E, Charles AL, Bender L, Baeza C, Pélissie M, Trensz P, Pflumio C, Kalish-Weindling M, Gény B, Schott R, Favret F, Pivot X, Hureau TJ, Pagano AF. Chemotherapy impairs skeletal muscle mitochondrial homeostasis in early breast cancer patients. J Cachexia Sarcopenia Muscle. 2022 Jun;13(3):1896-1907. doi: 10.1002/jcsm.12991. Epub 2022 Apr 4. PMID: 35373507.

Authors showed increased activity of AMPK in C8 enriched model. It could be more supportive for the oxidative capacity of the mitochondria if they have included an experiment to measure ACC phosphorylation in C8 positive and negative model.

à Thank you for this comment. Indeed, the measurement of ACC phosphorylation could be a supportive proof of AMPK activation. In our case, we choose to focus on mitochondrial biogenesis pathway, and as AMPK activation increase mitochondrial biogenesis throughout PGC1a, we focus on this evidence, because ACC is more involved in fatty acid storage than in mitochondrial biogenesis. However, we take note of this suggestion to improve our future research.

Keep using scientific format of et al throughout the manuscript. Correct Watt and al, Takikawa and al, and Loyd and al as watt et al and so.

à Thanks, we made the correction.

Sentences, “To understand the way involved in the improvement of endurance capacity and spontaneous activity, we explored the metabolic phenotype of skeletal muscle” and “As carbohydrate metabolism is the first way of ATP production, carbohydrate consumption before and during exercise is often recommended to improve performance” are not clear. What is meant by way? Could you please use appropriate word?

à We corrected “way” by “metabolic pathway” to be more accurate.

Provide proper label for WB images in figure 7.

à The WB images have been improved.

There is no housekeeping gene as a control for WB experiments. What is relative to control and how do you calculate it? Since t-AMPK expression is higher in C8- rich model, we can’t use t-AMPK as a control.

à Thank you for your comment. Even if single protein normalization is still widely used to normalize proteins of interest signals, it is now well admitted that total protein normalization is a more consistent loading control method. Indeed, please see below all the different studies/guidelines that suggest the use of Ponceau S (or total protein quantification) as an effective loading control method.

https://www.ncbi.nlm.nih.gov/pmc/articles/PMC3809032/

https://www.ncbi.nlm.nih.gov/pmc/articles/PMC6810642/

https://pubmed.ncbi.nlm.nih.gov/30914243/

https://pubmed.ncbi.nlm.nih.gov/20206115/

For the AMPK, a ratio was calculated for each sample (p-AMPK /t-AMPK) in order to reveal the activation of AMPK. T-AMPK was not used as a control, it was used to calculate the ratio, and the normalization has been done with the Ponceau.

Round 2

Reviewer 1 Report

Thank you for making the requested changes. I have no further edits.